# Copayment mechanism in selected districts of Uganda: Availability, market share and price of quality assured artemisinin-based combination therapies in private drug outlets

**Moses Ocan**[1]*, **Winnie Nambatya**[2], **Caroline Otike**[3], **Loyce Nakalembe**[4], **Sam Nsobya**[5]

1 Department of Pharmacology & Therapeutics, College of Health Sciences, Makerere University, Kampala, Uganda, 2 Department of Clinical Pharmacy, College of Health Sciences, Makerere University, Kampala, Uganda, 3 Data Department, Joint Clinical Research Centre, Lubowa, Kampala, Uganda, 4 Department of Pharmacology, Soroti University, Soroti, Uganda, 5 Department of Pathology, College of Health Sciences, Makerere University, Kampala, Uganda

* ocanmoses@gmail.com

## Abstract

### Background

Malaria remains one of the leading causes of morbidity, and mortality in Uganda. A large proportion of malaria symptomatic patients seek healthcare in private sector. However, availability and affordability are major barriers to access to effective treatment. The private sector copayment mechanism in Uganda aims to increase availability and affordability of antimalarial agents. Our study assessed availability, price, and market share of quality assured artemisinin-based combination therapies (QAACTs) in private drug outlets in selected districts during the implementation of copayment mechanism.

### Methods

This was a cross-sectional survey of anti-malarial agents in private drug outlets in in selected moderate-to-high (Tororo, and Apac districts) and low (Kabale and Mbarara districts) malaria transmission settings. Following the World Health Organization/Health Action International (WHO/HAI) criteria, an audit of the antimalarial agents was done using a checklist to determine availability, price, and market share of QAACTs. Data were entered in Epi-data and analyzed in STATA *ver* 14.0 at 95% confidence level.

### Results

A total of twenty-eight (28) private drug outlets (pharmacies and drug shops) were included in the survey. One in seven (20/144: 95%CI: 9.1, 20.6) of the antimalarial agents in private drug outlets were quality assured artemisinin-based combination therapies (QAACT). Artemether-lumefantrine (AL), 8.9% (11/124) and Artesunate-Amodiaquine (AQ), 7.3% (9/124) were the only QAACTs present in the drug outlets at the time of the survey. The majority, 86.1%% (124/144) of antimalarial agents present in stock in the drug outlets were

**Data Availability Statement:** All relevant data are within the manuscript and its Supporting Information files.

**Funding:** This study is part of the EDCTP2 programme supported by the European Union (TMA2019CDF-2662-Pfkelch13 emergence). The views and opinions of authors expressed herein do not necessarily state or reflect those of EDCTP. The funders had no role in study design, data collection and analysis, decision to publish, or preparation of the manuscript.

**Competing interests:** The authors have declared that no competing interest exists.

artemisinin based. The most common, 38.9% (56/144) ACT in the drug outlets was Dihydroartemisinin-Piperaquine (DHP). Most, 69.4% (100/144) of the antimalarial agents were in high malaria transmission settings. The cost of ACT antimalarial agents is high in the country, USD 1.4 (Artemether-Lumefantrine, AL), USD 2.4 (Dihydroartemisinin-Piperaquine, DP), the first line and second-line agents respectively for treatment of uncomplicated malaria in Uganda. There was a statistically significant difference between the dispensing price of 'Green leaf' ACTs (QAACT) and the recommended price ($p$<0.001). Predictors of availability of QAACT in private drug outlets include pharmacy drug outlet (aPR:0.4; 95%CI: 0.2, 0.9) and dispensing price more than 3000UGX (USD 0.83) (aPR: 0.4, 95%CI: 0.1, 0.51).

## Conclusion

Quality assured artemisinin-based combination therapies (QAACTs) are not common in private drug outlets in selected districts in Uganda. All the drug outlets had at least one ACT antimalarial agent present on the day of the survey. The dispensing price of QAACTs was significantly higher than the recommended markup price. There is need for awareness creation, surveillance, and monitoring of the implementation of Copayment mechanism in the country.

## Introduction

The World Health Organization (WHO) reported an overall increase in malaria burden globally of up to two million cases in 2020–2021 with most of the increase occurring in the WHO Africa region [1]. Malaria remains a disease of public health concern in Uganda and is responsible for up to 40 percent of all outpatient visits, 25 percent of hospital admissions and 14 percent of all hospital deaths [2, 3]. Although there has been remarkable decrease in malaria prevalence in Uganda, these gains remain fragile due to inconsistency in implementation of the malaria vector control measures [4]. Recent studies in Uganda [5, 6] have reported a resurgence in malaria burden to pre-indoor residual spraying (IRS) levels despite sustained IRS across sentinel surveillance sites. Therefore, despite implementation of vector control measures, malaria cases continue to rise after initial reduction due to IRS in historically high burden districts in North-Eastern Uganda [5–7].

The healthcare system in Uganda like most other low-and middle-income countries (LMICs) is faced with multiple challenges including inadequate human resource and limited funding [8]. The unreliable supply coupled with high costs of antimalarial agents in health facilities common in most LMICs potentially affect malaria treatment [9, 10]. In Uganda, this is likely to be worsened by the high burden of malaria.

In most low-and middle-income countries (LMICs), over 60% of the population first seek treatment in the private sector [11]. A study by Ocan *et al.*, [12] reported over half of the population in northern Uganda access antimalarial agents over the counter in private drug outlets. The sector thus plays an integral role in the fight against malaria [13]. Patients seeking malaria treatment in the private sector are however faced with multiple challenges including high cost, unreliable supply, and substandard antimalarial agents [10, 14]. In most malaria affected countries price remains the leading contributor to limited access to antimalarial agents [10, 15]. These potentially affects access to appropriate malaria treatment which could predispose

patients to adverse treatment outcomes including exacerbation of the current reported resistance to first-line antimalarial agents in the country.

One of the key factors influencing access to appropriate malaria treatment is pricing of antimalarial agents [10]. The Affordable medicines facility for malaria (AMFm) is a financing mechanism incorporating three elements; price reductions through negotiations with manufacturers of quality-assured ACTs (QAACTs), a buyer subsidy to the participating manufacturers, and interventions to support AMFm implementation and promotion of appropriate antimalarial use [16]. Affordable Medicines Facility-malaria (AMFm), funded by three donors (UNITAID, the UK government and the Bill & Melinda Gates Foundation) was renamed as the private sector co-payment mechanism and integrated into the existing Global Fund grants [17]. The AMFm initiative is intended to improve availability, affordability, and access to quality assured artemisinin-based combination therapies (QAACTs) in the private sector [18]. Under the AMFm, public and private importers pay manufacturers a subsidized price between $0.005 and $ 0.22 per treatment course, representing 1–20% of the manufacturer price [19]. The balance is then paid to the manufacturer via the co-payment fund [19]. Uganda was one of the countries where the initiative was first piloted and subsequently implemented the Copayment mechanism following the AMFm transition period in 2013 [18].

Negotiations with manufacturers under the AMFm led to reductions in manufacturer price for sales to private-for-profit buyers of up to 29–78% depending on package size, bringing prices down to around the levels paid by public purchasers [20]. This helped improve availability, affordability, and access to QAACTs in the AMFm pilot countries. In Uganda the AMFm was replaced by Copayment mechanism following the end of the pilot period [18]. The approved quality assured ACTs in Uganda include Artemether/lumefantrine, Artesunate/Amodiaquine, Dihydroartemisinin/piperaquine, and Artesunate/Pyronaridine [21, 22]. Uganda is currently implementing 70–80% subsidy for QAACTs [18]. However, the impact of the Copayment mechanism in the private sector remains unclear. Therefore, this study assessed availability, price, and market share of quality assured artemisinin-based combination therapies (QAACT) in private drug outlets in selected districts in Uganda following implementation of Copayment mechanism.

## Methods

### Study design and setting

This was a cross-sectional study conducted in selected districts in low (Kabale and Mbarara districts) and moderate -to-high (Apac and Tororo districts) malaria transmission settings. We used the Ministry of health stratification of malaria transmission intensity across the country to identify areas categorized as having moderate-to-high and low malaria transmission. Malaria transmission in the country is stratified into four levels based on malaria cases, 'high' (> 499 cases per 1000 population/year), 'moderate' (200 to 499 cases per 1000 population per year), 'low' level 2 (50 to 199 cases per 1000 population per year), 'very low' level 1 (1–49 cases per 1000 population per year) [23]. Tororo and Apac are districts with moderate-to-high malaria transmission while Mbarara and Kabale are low malaria transmission areas [4, 23]. Data on antimalarial agents present in the private drug outlets on the day of the survey was collected.

### Sample size, drug outlet selection and sampling

The WHO/HAI methodology was used for collection of data on availability and price of antimalarial agents in this study [24]. In each district (Tororo, Apac, Mbarara and Kabale), a comprehensive list of the available private drug outlets was compiled using the National drug

authority (NDA) register of drug outlets. Additionally, a census was taken to identify all functional private drug outlets (pharmacies and drug shops) in each district prior to data collection. From the census and review of drug outlets listed in the NDA register, each of the available drug outlet in the survey districts were considered for inclusion in the survey. The drug outlet which reported stocking antimalarial agents were noted and the batches of different individual antimalarial agents were used to determine if the agent would be included in the survey. Data on the antimalarial agent was collected whenever a batch that has not been encountered before was found in the drug outlet. Data on each batch was collected once in a total of 28 different private drug outlets. Private drug outlets in this study are defined as for-profit licensed establishments that dispense medicines. Two research assistants, a pharmacist (KJ) and nurse (RK) conducted the antimalarial drug audit in the private drug outlets. Each of the research assistants separately visited different drug outlets in the study districts. At each drug outlet prior to data collection the research assistants introduced themselves and explained the study and provided approval letters from the Ethics committee, UNCST and local authorities. The research assistants obtained a written informed consent from the pharmacists/dispensers in each pharmacy prior to data collection which was done between 15th June 2021- 21st December 2021. In each study district, the research assistants moved from one drug outlet to the next until all the identified private drug outlets were covered.

## Data collection

In each drug outlet, the dispenser/pharmacist was requested to show the research assistants all the antimalarial agents available in stock on the day of the survey. Additionally, a stock card and dispensing logs were also obtained and screened. An audit of the antimalarial agents that were provided, stock cards and dispensing logs was then done using a data collection checklist. The tool was pre-tested in drug outlets in Kampala city and validated. Data were collected for each identified anti-malarial agent in the drug outlet. The WHO list of antimalarial agents was used in identification of antimalarial agents present in stock during the day of the survey. Additionally, the Ministry of Health list of quality assured artemisinin-based agents in the country was also used in identifying QAACTs in the drug outlets. Data were collected on, drug outlet type, generic name, brand name, active ingredients and strengths, formulation, quality assured ACT (having a green leaf logo), non-quality assured ACT (without a green leaf logo), manufacturer and country of manufacture, type of antimalarial (artemisinin or non-artemisinin based), location of drug outlet (transmission setting), and package size. The price of the antimalarial agents was collected from the dispensing logs and the dispenser/pharmacist.

During each data collection day, the lead researcher reviewed all the data collection forms and visited 20% of all the drug outlets for quality control checks. Any discrepancies in the data collection tool were rectified through discussion with the data collectors and consensus.

## Data management and analysis

In this study, ACT antimalarial agents were classified into two categories: i) quality assured ACTs (QAACTs) are ACTs that comply with the Global Fund's ACT chemical regulation and have a green leaf logo printed on its label and ii) non-quality assured ACTs (non-QAACTs) which are ACTs without the green leaf logo on their labels (S1 Dataset). The Quality-assured ACTs were further divided into two categories: i) quality-assured first-line ACT (Artemether/ Lumefantrine) and ii) quality-assured alternative first-line ACT (Artesunate/Amodiaquine) as per the national malaria treatment guidelines [25]. Antimalarial availability was measured as the proportion of outlets with at least one anti-malarial agent present on the day of data collection (survey). Additionally, in each private drug outlet the proportion of specific anti-malarial

categories present in stock on the day of the survey was also calculated. The prices of antimalarial agents in the drug outlets were converted to US dollars using the average annual exchange rate by the Bank of Uganda in 2021. The antimalarial price data at the time of data collection is reported in terms of adult equivalent doses, defined as the amount needed to treat a 60 kg adult [26].

Data were entered in Epi-Data *version* 4.2.0 and exported to STATA *version* 14.0 for cleaning and analysis. Antimalarial agents were summarized using frequencies and proportions, prevalence of antimalarial agents was also determined using proportions. Distribution of antimalarial agents across malaria transmission settings and type of drug outlet was compared using fisher's exact test of independence. Price of antimalarial agents was summarized using median and interquartile range. The factors associated with availability of quality assured artemisinin-based combination therapies in private drug outlets was assessed using modified Poisson regression at bivariate level to establish the crude relationship between the predictors and the outcome variable, and multivariate levels to assess the adjusted relationship between the predictors and the outcome variables. The factors that had P-value of 0.2 and below from the bivariate analysis were included in the multivariable regression analysis. The Ministry of Health set dispensing price of 'Green leaf' (QAACT) artemether-lumefantrine (20/120mg), the first line agent for treatment of uncomplicated malaria is USD 0.83 (UGX 3000) which was used in regression analysis. Statistical significance was determined at 5% level of significance.

## Results

### Descriptive results

A total of twenty-eight (28) drug outlets were included in the survey of which majority, 78.6% (22/28) were pharmacies. All the drug outlets in the selected study districts (Apac, Tororo, Mbarara and Kabale) had at least one antimalarial agent present in stock during the day of the survey. Of the 28 drug outlets visited in this study, majority 82.1% (23/28) had a minimum of 5 different kinds of antimalarial agents present on the day of the survey. Most, 69.4% (100/144) of the antimalarial agents were in high malaria transmission settings. The majority, 86% (124/144) of the antimalarial agents were tablet formulations. Twelve, 8.3% (12/144) of the artemisinin based antimalarial agents in the drug outlets were monotherapies (Table 1).

### Antimalarial agents present in the drug outlets in selected districts on the day of the survey

A total of 144 antimalarial agents were present in the drug outlets on the day of the survey of which majority, 86.1% (124/144) were artemisinin-based agents. Of the Artemisinin based combination therapies, 61.1% (88/144) were in high malaria transmission settings and 68.8% (99/144) were found in pharmacies. For non-artemisinin based antimalarial agents, 8.3% (12/144) were in high malaria transmission setting and 11.1% (16/144) were in pharmacies.

Of the artemisinin-based agents, majority 90.3% (112/124) were Artemisinin-based combination therapies (ACTs). Artesunate was the most common artemisinin monotherapy, 8.9% (11/124). All the artemisinin monotherapies 9.7% (12/124) were injectables (Artemether and Artesunate). Over a third 35.4% (51/144) of the antimalarial agents were first-line ACTs (Artemether-lumefantrine, 29.2%: 42/144 and Artesunate-amodiaquine, 6.3%: 9/144). Most first-line ACTs, 26.4% (38/144), 25.7% (37/144) were in high malaria transmission setting and pharmacies respectively. Of the ACTs 70.2% (87/124), 66.9% (83/124) were present in pharmacies and high malaria transmission settings respectively. There was no statistically significant difference in the presence of ACTs in pharmacies and drug shops (*p* = 0.318). However, there

**Table 1. Antimalarial agents (N = 144) in drug outlets in moderate-to-high and low malaria transmission settings in Uganda, June-December 2021.**

| Antimalarial category | Description | Overall n (%) | Low malaria transmission n (%) | High malaria transmission n (%) | p-value |
|---|---|---|---|---|---|
| Artemisinin monotherapies | Artesunate | 11 (7.6) | 6 (4.1) | 5 (3.5) | 1.00 |
| | Artemether | 1 (0.7) | 1 (0.7) | 0 | |
| Non-Artemisinin based monotherapies | Chloroquine | 2 (1.4) | 1 (0.7) | 1 (0.7) | 1.00 |
| | Quinine | 8 (5.6) | 3 (2.1) | 5 (3.5) | |
| | Mefloquine | 1 (0.7) | 0 | 1 (0.7) | |
| Artemisinin-based combination therapies (ACTs) | Artemether/lumefantrine | 42 (29.1) | 13 (9) | 29 (20.1) | 0.213 |
| | Artesunate/Amodiaquine | 9 (6.3) | 0 | 9 (6.3) | |
| | Dihydroartemisinin/ Piperaquine | 56 (38.9) | 15 (10.4) | 41 (28.5) | |
| | Artesunate /Mefloquine | 2 (1.4) | 1 (0.7) | 1 (0.7) | |
| | Artesunate/Pyronaridine | 3 (2.1) | 0 | 3 (2.1) | |
| Non-Artemisinin based combination therapies | Atovaquone/Proguanil | 2 (1.4) | 1 (0.7) | 1 (0.7) | 1.00 |
| | Sulfadoxine/Pyrimethamine | 7 (4.9) | 3 (2.1) | 4 (2.8) | |
| Quality Assured ACTs ('Green leaf ACTs | Artemether/lumefantrine | 11 (7.6) | 3 (2.1) | 8 (5.5) | 0.218 |
| | Artesunate/Amodiaquine | 9 (6.3) | 0 | 9 (6.3) | |
| Expired antimalarial agents | Yes | 4 (2.8) | 1 (0.7) | 3 (2.1) | 1.00 |
| | No | 140 (97.2) | 43 (29.9) | 97 (67.3) | |
| Formulation | Powder | 18 (12.5) | 10 (6.9) | 8 (5.5) | 0.039 |
| | Syrup | 2 (1.4) | 0 | 2 (1.4) | |
| | Tablets | 124 (86.1) | 34 (23.6) | 90 (62.5) | |

ACTs: Artemisinin-based Combination therapies; n = sample size; %: percentage

was a statistically significant difference in the distribution of ACTs based on setting ($p = 0.03$). Four batches, 2.8% of the Artemisinin based antimalarial agents available in stock in the drug outlets had passed their expiry date. Of the expired agents, three were artemisinin-based while one was a non-artemisinin-based agent (Table 1).

## Availability of quality assured artemisinin combination therapies (QAACTs) in selected districts

One in seven (20/144: 95%CI: 9.1, 20.6) of the antimalarial agents in private drug outlets were quality assured artemisinin-based combination therapies ('Green leaf ACTs'; QAACT). Most, 9.7% (12/124) of the QAACTs were in pharmacies. The QAACTs were common, 13.7% (17/124) in drug outlets in high malaria transmission settings. Artemether-lumefantrine (AL), 8.9% (11/124) and Artesunate-Amodiaquine (AQ), 7.3% (9/124) were the only QAACTs present in drug outlets on the day of the survey. All the Artesunate/Amodiaquine agents in drug outlets were 'Green leaf' ACTs (Table 1).

## Price of antimalarial agents in drug outlets in selected districts of Uganda

Arterolane maleate/piperaquine phosphate combination had highest median price among the antimalarial combination agents, 4.9 USD. The median price for Artemether/lumefantrine (AL), the first line agent for treatment of uncomplicated malaria in Uganda was USD 1.4.

**Table 2. Median price of oral antimalarial agents in drug outlets in selected districts in Uganda June-December 2021.**

| S/N | Generic name | Median price (USD) | IQR (USD) |
|-----|--------------|--------------------|-----------|
| 1. | Artemether/Lumefantrine | 1.4 | 0.9, 1.9 |
| 2. | Arterolane maleate/piperaquine Phosphate | 4.9 | 2.4, 5.6 |
| 3. | Artesunate + Amodiaquine | 0.8 | 0.7, 1.3 |
| 4. | Artesunate/Mefloquine | 2.8 | 2, 4.2 |
| 5. | Chloroquine Phosphate | 0.6 | 0.5, 0.8 |
| 6. | Dihydroartemisinin/Piperaquine | 3.2 | 2.7, 4.2 |
| 7. | Mefloquine | 1.1 | 0.7, 2.2 |
| 8. | Quinine sulphate | 0.4 | 0.3, 0.4 |
| 9. | Sulfadoxine/Pyrimethamine | 0.6 | 0.1, 0.4 |
| 10. | Atovaquone/Proguanil | 1.9 | 1.8, 2.0 |
| 11. | Artesunate/Pyronaridine | 2.2 | 0.1, 2.8 |
| 12. | Arterolane maleate/piperaquine Phosphate | 4.9 | 2.4, 5.6 |

USD: US dollar, IQR: Interquartile range

Dihydroartemisinin/Piperaquine (DP), the second-line agent for treatment of uncomplicate malaria in Uganda cost was sold at a median price of USD 3.2. Artesunate + Amodiaquine (AS +AQ), the second-line alternative agent for uncomplicate treatment of uncomplicate malaria in Uganda the median price was USD 0.8 (Table 2).

## Price mark-ups on quality assured artemisinin-based combination therapies (QAACTs) in drug outlets in selected districts in Uganda

The median dispensing price for adult dose of quality assured (QAACT: 'Green leaf') first-line ACT for uncomplicated malaria, artemether-lumefantrine (AL) (20/120mg) was USD 0.98, which is higher than the recommended retail price by 10.2%. The median dispensing price for quality assured Artesunate/Amodiaquine (AQ) (25/67.5mg) was USD 0.7 which is higher than the recommended retail price by 57%. There was a statistically significant difference between the dispensing price of the 'Green leaf' ACTs and the recommended retail price ($p<0.001$).

## Predictors of availability of quality assured artemisinin-based combination therapies (QAACT) in private drug outlets in selected districts of Uganda

At bivariate regression analysis, the factors associated with availability of QAACT in drug outlets include pharmacy (cPR: 0.48; 95%CI: 0.2, 1.1; $p = 0.07$), low malaria transmission (cPR:1.83; 95%CI: 0.11, 0.5; $p = 0.35$), and dispensing price > 3000UGX (USD = 0.83) (cPR: 0.23; 95%CI: 0.11, 0.5; $p<0.000$).

At multivariable regression analysis, the predictors of availability of QAACT in drug outlets include pharmacy (aPR:0.4; 95%CI: 0.2, 0.9; $p = 0.019$) and dispensing price more than 3000UGX (USD 0.83) (aPR: 0.4, 95%CI: 0.1, 0.51; $p<0.000$) (Table 3).

## Discussion

The study found at least one antimalarial agent present in all drug outlets during the day of the survey. Additionally, Artemisinin-based combination therapies (ACTs) were the most common antimalarial agents in the drug outlets. This is unlike the findings of a previous study by O'Connell et al., [27] done in six sub-Saharan African countries that reported a low prevalence

**Table 3. Predictors of availability of quality assured ACTs in drug outlets in selected districts in Uganda, June-December 2021.**

| Characteristic | Description | cPR (95%CI) | aPR (95%CI) |
|---|---|---|---|
| Drug outlet | Drug shop | 1.0 | 1.0 |
| | Pharmacy | 0.48 (0.2, 1.1) | 0.4 (0.2, 0.9) |
| Malaria transmission setting | High | 1.0 | - |
| | Low | 1.83 (0.51, 6.51) | - |
| Price category | <3000 UGX (USD 0.83) | 1.0 | 1.0 |
| | >3000 UGX (USD 0.83) | 0.23 (0.11, 0.5) | 0.23 (0.1, 0.51) |

cPR: crude prevalence ration, aPR: adjusted prevalence ration, CI: Confidence interval, UGX: Ugandan Shillings

of ACTs in private sector. The implementation of Affordable medicines facility for malaria (AMFm) program could have influenced the difference in the findings of the two studies. A study by ACTwatch group *et al.*, [18], reported an increase on availability of ACTs following implementation of the AMFm-malaria program. The finding of our study is an indicator of wide availability of Artemisinin-based combination therapies (ACTs), the current cornerstone for malaria treatment in private drug outlets.

The immediate post AMFm pilot period was characterized by increases in the availability of QAACTs in the private sector in Uganda and in all other countries where it was implemented [18, 19]. However, our study found a low prevalence of QAACTs in private drug outlets after over ten years since the introduction of copayment mechanism following expiry of AMFm pilot period. This is like findings of a study by O'Connell et al., [27]. Furthermore, the ACTs in pharmacies were 52% less likely to be quality assured. The low availability of QAACTs in private drug outlets highlights the challenges of implementation of copayment mechanism especially lack of supporting interventions like training of private sector healthcare personnel and behavioral change communication [18, 19].

The study did not find oral artemisinin-based monotherapies in the drug outlets. The injectable artemisinin monotherapies (Artesunate and Artemether) were however common. According to the National malaria case management guidelines, injectable artemisinin agents are indicated in the treatment of severe malaria and are to be used alongside oral ACTs. The use of oral artemisinin monotherapies in malaria treatment was reported in Southeast Asia as one of the drivers of artemisinin resistance development [28]. Our findings are like those of a previous study by ACTwatch et al., [18] and O'Connell et al., [27] who reported not finding artemisinin-based monotherapies in sub-Saharan Africa. The objectives of the AMFm were to make the ACTs more available, accessible and to crowd out artemisinin monotherapies [18]. The findings of our study could thus be due to the emphasis placed on the Artemisinin based combination therapies through the AMFm and Copayment mechanism in the country. In addition, the absence of artemisinin monotherapies in Uganda could be due to the World Health Organization (WHO) recommendation to malaria affected countries to ban importation and manufacture of these agents.

The study found non-artemisinin antimalarial agents in the drug outlets across the country. This is like the findings of a study by O'Connell et al., [27] which reported high prevalence of non-artemisinin agents in sub-Saharan African countries. Our study found, chloroquine, quinine and Sulphadoxine/Pyrimethamine (SP) as common non-artemisinin agents in drug outlets. Presence of non-artemisinin therapies in the private drug outlets could be due to their perceived effectiveness and familiarity from previous use coupled with provider behavior and beliefs [12, 29]. Sulphadoxine/Pyrimethamine (SP) is recommended for use in prophylaxis

against malaria in pregnancy [30]. Additionally, Quinine is a second line agent in treatment of severe malaria in the country [25]. However, the use of chloroquine was discontinued in 2006 [31] due to widespread resistance. The continued presence of chloroquine in private drug outlets is an indicator of access and use by communities in management of symptoms of malaria. Recent studies in Uganda have reported persistence of *P. falciparum* resistance against chloroquine over two decades after discontinuation of its use in the country [32]. Therefore, continued use of chloroquine by communities is likely to lead to inadequate treatment and increased risk of unwanted malaria treatment outcomes.

Dihydroartemisinin/Piperaquine (DP) and Artemether/lumefantrine (AL), the two most popular ACTs had a high median dispensing price for a full adult dose ranging from USD 1.4–2.4. The median price of the subsidized ACTs ('Green leaf ACTs') was also significantly higher than the non-subsidized agents. The ACTs that were priced more than USD 0.83 were 77% less likely to be quality assured. This is similar to the prices reported in a study by O'Connell et al., [27] done in six malaria endemic countries in sub-Saharan Africa. A previous study by Tougher et al., [19] also found no significant change in ACT prices following implementation of AMFm pilot program in Uganda. In a country where majority of the rural communities live under USD 1.0 a day, the reported prices of ACTs are potentially prohibitive. The reported high cost of ACTs potentially hinders access to appropriate and effective malaria treatment in the country. Recent studies in Uganda have reported high over-the-counter access of antimalarial agents which coupled with the high cost of ACTs could drive inappropriate use [12, 33]. With the emerging artemisinin resistance, inappropriate use of ACTs is likely to exacerbate malaria parasite resistance in the country [34]. The high cost of quality assured ACTs in the private sector despite government subsidy highlights the challenge of implementation of government programs especially in LMICs. There is thus need for regular monitoring of the implementation of copayment mechanism for improvement of availability and access to quality assured ACTs in the private sector.

The study had some limitations, the overt method used is likely to be affected by the pharmacists not reporting the actual practice in the pharmacy. However, the additional review of stock cards and drug dispensing logs in the pharmacies and drug shops helped collaborate the findings of the study. Additionally, the study did not collect data from all districts in the country however inclusion of districts with moderate-to-high and low malaria transmission helped in ensuring representation of settings with diverse malaria transmission intensity in the country. However, the findings of our study may not reflect the effects of copayment program in the entire country.

## Conclusion

The results of our study indicate low prevalence of QAACTs in private drug outlets in selected districts in Uganda despite the implementation of private sector copayment mechanism. Prices of both QAACTs and non-QAACTs remain high in private drug outlets (pharmacies and drug shops) in selected districts despite the subsidies on QAACTs. There is need for consistent implementation of Copayment mechanism supporting interventions to help create demand for QAACTs among private sector health providers and the population. The Ministry of Health needs to conduct a review of the copayment mechanism to establish barriers to its implementation in the country.

## Supporting information

**S1 Dataset. Minimal dataset.**
(XLS)

## Acknowledgments

We acknowledge Mr. Tayebwa Mordecai and Ms. Joanita Birungi for managing and coordinating field data collection. We are grateful to the research assistants, Ms. Ruth Kokusiima, Ms. Kadesemba Phoenah, Mr. Olwortho Wilfred, and Mr. Kato Henry for the work done during the field data collection. All authors read and approved the final manuscript.

## Author Contributions

**Conceptualization:** Moses Ocan, Sam Nsobya.

**Data curation:** Caroline Otike.

**Formal analysis:** Caroline Otike.

**Investigation:** Moses Ocan.

**Methodology:** Moses Ocan.

**Supervision:** Loyce Nakalembe, Sam Nsobya.

**Validation:** Winnie Nambatya.

**Writing – original draft:** Moses Ocan.

**Writing – review & editing:** Moses Ocan, Winnie Nambatya, Loyce Nakalembe, Sam Nsobya.

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
