## [Editor Report · Decision Letter 0]

29 Nov 2023

PONE-D-23-37429Availability, market share and price of quality assured artemisinin-based combination therapies in private drug outlets after over a decade of Copayment mechanism in Uganda.PLOS ONE

Dear Dr. Ocan,

Thank you for submitting your manuscript to PLOS ONE. After careful consideration at the initial review, we feel that it has merit but does not fully meet PLOS ONE’s publication criteria as it currently stands.  There are important methodological issues that we would like you to address before it can be circulated for review. Therefore, we invite you to submit a revised version of the manuscript that addresses the points raised during the initial review process. Please submit your revised manuscript within Jan 13 2024 11:59PM.

We look forward to receiving your revised manuscript.

Kind regards,

Bosco Bekiita Agaba, PhD

Academic Editor

PLOS ONE

The study “Availability, market share and price of quality assured artemisinin-based combination therapies in private drug outlets after over a decade of Copayment mechanism in Uganda” aimed to assess the availability, price, and market share of quality assured artemisinin-based combination therapies (QAACT) in the private drug outlets in low and high malaria transmission settings in Uganda” The authors conducted the study using a cross-sectional design covering and drawing their conclusions based on twenty-eight (28) drug outlets covered.

Overall, it is a useful study as it brings out data that could inform case management but also feeds into the overall ACTs resistance management plan.

General comments

The authors report to have found and included only 28 outlets (drug shops and pharmacies) located in 4 districts covered by the survey. The authors indicate to have used the NDA list of all licensed outlets, conducted a census of all the outlets and enrolled all of them (drug shops and pharmacies) in these districts. Could the authors cross-check if this information is correct? that there are only 28 licensed outlets (pharmacies and drug shops) across the 4 districts?

ACTs procured under the subsidized/copayment mechanism are not distributed to the 4 districts covered by the survey alone but to the entire country of 140 + districts). In my opinion, it be a source of bias to base on four purposely/conveniently picked districts to report and make conclusion on the impact of a decade long program. What do the authors think?

The supply chain system (first line buyers, retailers etc) through which these subsidized/copayment ACTs are distributed entirely relies on market forces, demand creation, etc. Because of these forces and dynamics, there are variations in the market share of these drugs between areas/regions within a country. Eg if you have more distributors moving to same direction you are likely to find more copaid drugs there compared to the rest. The best suitable method to answer the question would therefore entail sampling of outlets in a way that is representative and spread across wide coverage of the country. What do the authors think?

The title of the study appears to be misleading vs the method used and data presented. Copayments/subsidized ACTs in most countries have been supported by donor countries majorly the Global Fund. These grants have time periods within which they expire. After grant expiry, it might take a minimum of 2-3 years to link to the new grant. Therefore, the assumption of 10 years implementing copayment is misleading as there are frequent lapses of non-implementation intervals in between- with the frequent interruption/intervals of non-implementation within the “decade” is the “decade” actually a “decade”

Specific comments

117-118: In all the countries where ACT copayments are being implemented, malaria transmission or endemicity level is not one of the criteria while making decisions around implementing the subsidized/copayments both under the current GF funded grants in Africa and the previously under the AMFm. The major target is to remove barriers to access, availability and affordability particularly in hard to reach and vulnerable populations that are at greatest risk of malaria infections regardless of transmission/endemicity level. The convenient selection of districts from low and high transmission doesn’t seem to add value in view of the key goals of copayments.

202-203: Survey covered 28 outlets- small size samples are characterized by wide confidence intervals. It is not clear whether the P-values obtained are not due to small size/numbers. The confidence intervals are not shown throughout the report. What do the authors think about the strength of this evidence in view of such important intervention?

Table 1: ACT Copayments are designed to achieve cost /price reduction and access, availability- it would be more informative if the table was stratified to bring out the key copayment aspects

110-111: There are several programmatic reports (unpublished) that have evaluated the effect of ACT copayment. It would be great to access and see what these reports say vs this study

117-118: malaria epidemiology is highly dynamic and transmission is highly heterogenous and this complicates epidemiological stratification into high, Moderate and low transmission based on parasite prevalence/slide positivity rates alone. Areas that were originally high are now low and vice versa and this phenomenon is happening in most countries in Africa. Accurate Epidemiological stratification is based on a composite factor that combines EIRs, parasite rates & positivity rates, etc. what parameters were based arrive at the transmission levels are reported? Is there data on the EIRs for these areas indicated?

122-126: were there discrepancies between the NDA list and the physical census?

139: HAI Write in full time first

Line 354: While the conclusion is strong, the evidence /data and the methods used appear sub-optimal to support it (for the reasons elaborated above)

1. The selection was purposively and conveniently selected- which could have introduced bias. There was no attempt to address this at analysis either through regression to adjust for it or by any other means

2. A sample of 28 outlets is certainly too small to draw conclusions on a decade long intervention that covers the entire country

3. Conducting a simple cross-sectional survey at the end of an important project such as this, would assume a decade long, contentious uninterrupted implementation of ACT copayment/subsided ACT project which would be rather misleading. There are about three incidents of 2-3 year periods of interrupted non-implementation. A cross sectional study certainly masks important information within the decade.

4. The study looked at outlets in high and low transmission, a criterion that is not very useful for the ACTS Copayments (access, cost, hard to reach, etc)

5. The epidemiological criteria for stratifying districts into low and high was not was not indicated

6. overall, the methods and the data generated and the evidence reported do not adequately support the conclusion for such an important important intervention.

---

## [Author Response · Author response to Decision Letter 0]

5 Dec 2023

RESPONSE TO REVIEWER’S COMMENTS PONE-D-23-37429

We are grateful for the comments from the reviewers of our manuscript. The comments are insightful and responding to them have helped improve the manuscript. Find below a summary of our responses to the comments raised on the manuscript by the reviewers.

Comment #1: The study “Availability, market share and price of quality assured artemisinin-based combination therapies in private drug outlets after over a decade of Copayment mechanism in Uganda” aimed to assess the availability, price, and market share of quality assured artemisinin-based combination therapies (QAACT) in the private drug outlets in low and high malaria transmission settings in Uganda” The authors conducted the study using a cross-sectional design covering and drawing their conclusions based on twenty-eight (28) drug outlets covered. Overall, it is a useful study as it brings out data that could inform case management but also feeds into the overall ACTs resistance management plan.

Response: Thanks for the appreciation. 

General comments

Comment #2: The authors report to have found and included only 28 outlets (drug shops and pharmacies) located in 4 districts covered by the survey. The authors indicate to have used the NDA list of all licensed outlets, conducted a census of all the outlets and enrolled all of them (drug shops and pharmacies) in these districts. Could the authors cross-check if this information is correct? that there are only 28 licensed outlets (pharmacies and drug shops) across the 4 districts?

Response: Thanks for the comments, it is not true that there are only 28 drug outlets in the four districts. As per the drug outlet register available at https://www.nda.or.ug/licensed-outlets/ (Accessed 30th November 2023) there are a total of 122 pharmacies in the four districts (Mbarara, 88; Kabale, 16; Tororo, 16 and Apac, 2). Mbarara was recently divided into a city and district, and we collected data in the city not the district. As per the website, Mbarara city has 45 pharmacies. From the census and review of drug outlets listed in the NDA register, each of the available drug outlet in the survey districts were considered for inclusion in the survey. The drug outlet which reported stocking antimalarial agents were noted and the batches of different individual antimalarial agents were used to determine if the agent would be included in the survey. Data on the antimalarial agent was collected whenever a batch that has not been encountered before was found in the drug outlet. Data on each batch was collected once in a total of 28 different private drug outlets. This has been clarified in the revised manuscript. 

Comment #3: ACTs procured under the subsidized/copayment mechanism are not distributed to the 4 districts covered by the survey alone but to the entire country of 140 + districts). In my opinion, it be a source of bias to base on four purposely/conveniently picked districts to report and make conclusion on the impact of a decade long program. What do the authors think?

Response: Thanks for the comment, true the subsidized ACTs are not distributed in only the 4 districts included in this survey. The WHO/HAI recommends a total of 30 drug outlets (public and private) from six survey areas for medicine availability and price studies (5 drug outlets per survey area). In Uganda, medicines are supplied free of charge to patients in public health facilities and thus we did not include public facilities in the survey areas. We had a total of 4 survey areas and 28 drug outlets (22 pharmacies and 6 drug shops). We included a total of 144 antimalarial agents in the survey. The four districts were purposely selected to represent the malaria transmission patterns of the country. Both low and high malaria transmission settings are included in this study which is more likely to represent the true picture of malaria situation in the country. Apac and Tororo districts represent are the districts with the highest malaria transmission in the country. Kabale and Mbarara districts are the districts with lowest malaria transmission in the country. While the districts were purposely selected, all the available drug outlets in each district were visited and considered for inclusion in the survey reducing the risk of bias. Our findings are thus a true reflection of situation of subsidized ACTs in the country. 

Comment #4: The supply chain system (first line buyers, retailers etc) through which these subsidized/copayment ACTs are distributed entirely relies on market forces, demand creation, etc. Because of these forces and dynamics, there are variations in the market share of these drugs between areas/regions within a country. Eg if you have more distributors moving to same direction you are likely to find more copaid drugs there compared to the rest. The best suitable method to answer the question would therefore entail sampling of outlets in a way that is representative and spread across wide coverage of the country. What do the authors think?

Response: Thanks for the comment, true the distribution of the subsidized ACTs is likely to follow market forces as suggested by the reviewer. In our study we assessed the subsidized ACTs in both low and high malaria transmission settings which represent settings of potentially high demand due to high malaria transmission and low demand due to low malaria transmission. We agree with the reviewer, and this was factored in our study with the inclusion of areas of high and low malaria transmission. However, from the findings of our study there was no significant difference in the distribution of QAACT by malaria transmission setting (p=0.35) which is in line with the reviewer’s comment. This is highlighted in the revised manuscript.

Comment #5: The title of the study appears to be misleading vs the method used and data presented. Copayments/subsidized ACTs in most countries have been supported by donor countries majorly the Global Fund. These grants have time periods within which they expire. After grant expiry, it might take a minimum of 2-3 years to link to the new grant. Therefore, the assumption of 10 years implementing copayment is misleading as there are frequent lapses of non-implementation intervals in between- with the frequent interruption/intervals of non-implementation within the “decade” is the “decade” actually a “decade”

Response: The Affordable Medicines Facility for malaria (AMFm) pilot program was launched in 2010 consisting of nine pilots in eight countries: Cambodia, Ghana, Kenya, Madagascar, Niger, Nigeria, Uganda, Tanzania mainland and Zanzibar (The Global Fund to Fight AIDS, Tuberculosis and Malaria, 2008; ACTwatch et al., 2017). The pilot phase of program was from 2010 to 2011. After the expiry of the pilot phase, the AMFm transitioned to the private sector copayment mechanism (CPM) in 2013 in six countries: Ghana, Kenya, Madagascar, Nigeria, Tanzania, and Uganda. The co-payment mechanism has been implemented in Uganda since 2013 albeit with various challenges. Although the program has was launched in 2013, its implementation has been faced with various challenges which have caused interruptions to the program. We have published separately demonstrating the various challenges this program has faced in the country. We have removed the word ‘after a decade’ in the title of our study as guided by the reviewer. 

Specific comments

Comment #6: 117-118: In all the countries where ACT copayments are being implemented, malaria transmission or endemicity level is not one of the criteria while making decisions around implementing the subsidized/copayments both under the current GF funded grants in Africa and the previously under the AMFm. The major target is to remove barriers to access, availability and affordability particularly in hard to reach and vulnerable populations that are at greatest risk of malaria infections regardless of transmission/endemicity level. The convenient selection of districts from low and high transmission doesn’t seem to add value in view of the key goals of copayments.

Response: The prevalence of malaria is not uniform across Uganda which potentially creates variation in demand for the antimalarial agents in the country. In our study we ensured that the low and high malaria transmission (burden) settings are represented. Since our study was focused on the private sector, the design helped ensure representation of the settings of potentially low and high demand for the subsidized ACTs. However, from the findings of our study there was no significant difference in the distribution of QAACT by malaria transmission setting (p=0.35) which is in line with the reviewer’s comment. This is highlighted in the revised manuscript.

Comment #7: 202-203: Survey covered 28 outlets- small size samples are characterized by wide confidence intervals. It is not clear whether the P-values obtained are not due to small size/numbers. The confidence intervals are not shown throughout the report. What do the authors think about the strength of this evidence in view of such important intervention?

Response: The WHO/HAI recommends a total of 30 drug outlets (public and private) from six survey areas for medicine availability and price studies (5 drug outlets per survey area). In Uganda, medicines are supplied free of charge to patients in public health facilities and thus we did not include public facilities. In all the districts, we sampled more than 5 drug outlets. We had a total of 4 survey areas and 28 drug outlets (22 pharmacies and 6 drug shops). We included a total of 144 antimalarial agents in the survey. We have also included confidence intervals to our estimates indicating good precision in our estimates as there are no wide confidence intervals. 

Reference:

WHO/HAI [Health action international/World Health Organization] Medicine prices: a new approach to measurement. 2003 Ed. Geneva, World Health Organization and Health Action International, 2003. 

Comment #8: Table 1: ACT Copayments are designed to achieve cost /price reduction and access, availability- it would be more informative if the table was stratified to bring out the key copayment aspects.

Response: Thanks for the comment. Our results cover availability and price which are the key aspects of the copayment mechanism. The results are provided under the following sections in the manuscript; ‘Price of antimalarial agents in drug outlets in Uganda’ and ‘Price mark-ups on quality assured artemisinin-based combination therapies (QAACTs) in drug outlets in Uganda’

Comment #9: 110-111: There are several programmatic reports (unpublished) that have evaluated the effect of ACT copayment. It would be great to access and see what these reports say vs this study

Response: Thanks for the comment, we have reviewed both published and unpublished work in writing and revising the manuscript. 

Comment #10: 117-118: malaria epidemiology is highly dynamic, and transmission is highly heterogenous, and this complicates epidemiological stratification into high, Moderate and low transmission based on parasite prevalence/slide positivity rates alone. Areas that were originally high are now low and vice versa and this phenomenon is happening in most countries in Africa. Accurate Epidemiological stratification is based on a composite factor that combines EIRs, parasite rates & positivity rates, etc. what parameters were based arrive at the transmission levels are reported? Is there data on the EIRs for these areas indicated?

Response: We used the Ministry of Health stratification of malaria transmission settings across the country to identify areas categorized as having high and low malaria transmission. Tororo and Apac districts are categorized as having high malaria transmission and Mbarara and Kabale are low malaria transmission areas. This information has been incorporated in the revised manuscript. 

Comment #11: 122-126: were there discrepancies between the NDA list and the physical census?

Response: Yes, most of the drug outlets listed on the NDA website are not found on ground. 

Comment #12: 139: HAI Write in full time first.

Response: Thanks for the comment, this was already provided in the abstract. 

HAI: Health Action International. This is provided in the revised manuscript. 

Comment #13: Line 354: While the conclusion is strong, the evidence /data and the methods used appear sub-optimal to support it (for the reasons elaborated above)

Response: Thanks, we have addressed the comments raised on the manuscript and believe that our findings support the conclusion and reflects the situation of copayment mechanism in the country. 

Comment #14: 1. The selection was purposively and conveniently selected- which could have introduced bias. There was no attempt to address this at analysis either through regression to adjust for it or by any other means

Response: Thanks for the comment, in the revised manuscript we have conducted regression analysis and the results have been incorporated in the revised manuscript. 

Comment #15: 2. A sample of 28 outlets is certainly too small to draw conclusions on a decade long intervention that covers the entire country. 

Response: Thanks for the comment. The WHO/HAI recommends a total of 30 drug outlets (public and private) from six survey areas for medicine availability and price studies (5 drug outlets per survey area). In Uganda, medicines are supplied free of charge to patients in public health facilities and thus we did not include public facilities in the survey areas. We had a total of 4 survey areas and 28 drug outlets (22 pharmacies and 6 drug shops). We included a total of 144 antimalarial agents in the survey. We have also included confidence intervals to our estimates indicating good precision in our estimates. The study found 13.8% (20/144: 95%CI: 9.1, 20.6) availability of quality assured artemisinin-based combination therapy (QAACT) in private drug outlets. We also conducted regression analysis and results are provided in the revised manuscript. The factors associated with availability of quality assured artemisinin-based combination therapy was assessed using modified Poisson regression at bivariate level to establish the crude relationship between the predictors and the outcome variable, and multivariate levels to assess the adjusted relationship between the predictors and the outcome variables. Factors with P-value of 0.2 and below at bivariate analysis were included in the multivariable regression analysis. Statistical significance was determined at 5% level of significance. The results of the regression analysis are presented, and all have narrow confidence intervals. 

At bivariate regression analysis, the factors associated with availability of QAACT in drug outlets include pharmacy (cPR: 0.48; 95%CI: 0.2, 1.1; p=0.07), low malaria transmission (cPR:1.83; 95%CI: 0.11, 0.5; p=0.35), and dispensing price > 3000UGX (USD=0.83) (cPR: 0.23; 95%CI: 0.11, 0.5; p=<0.000).

Multivariable regression analysis, the predictors of availability of QAACT in drug outlets include pharmacy (aPR:0.4; 95%CI: 0.2, 0.9) and having dispensing price more than 3000UGX (USD 0.83) (aPR: 0.4, 95%CI: 0.1, 0.51) (Table 3). The ACTs that were priced more than USD 0.83 were 77% less likely to be quality assured. The ACTs in pharmacies were 52% less likely to be quality assured. 

Table 3: Predictors of availability of quality assured ACTs in drug outlets 

Characteristic Description cPR (95%CI) aPR (95%CI)

Drug outlet Drug shop 1.0 1.0

 Pharmacy 0.48 (0.2, 1.1) 0.4 (0.2, 0.9)

Malaria transmission setting High 1.0 -

 Low 1.83 (0.51, 6.51) -

Price category <3000 UGX (USD 0.83) 1.0 1.0

 >3000 UGX (USD 0.83) 0.23 (0.11, 0.5) 0.23 (0.1, 0.51)

cPR: crude prevalence ration, aPR: adjusted prevalence ration, CI: Confidence interval, UGX: Ugandan Shillings 

Comment #16: 3. Conducting a simple cross-sectional survey at the end of an important project such as this, would assume a decade long, contentious uninterrupted implementation of ACT copayment/subsided ACT project which would be rather misleading. There are about three incidents of 2-3 year periods of interrupted non-implementation. A cross sectional study certainly masks important information within the decade.

Response: Thanks for the comment, information on the interruptions to the implementation of copayment mechanism in Uganda is not readily available. We also followed the standard criteria by WHO/HAI for assessment of availability and price of medicines. However, we have adjusted the title of the manuscript to remove the word ‘decade’. 

Comment #17: 4. The study looked at outlets in high and low transmission, a criterion that is not very useful for the ACTS Copayments (access, cost, hard to reach, etc)

Response: The use of low and high malaria transmission setting was to ensure representation of areas of potentially varied demand for antimalarial agents. This is especially the case due to the influence of market forces of demand and supply in the private sector. However, from the findings of our study there was no significant difference in the distribution of QAACT by malaria transmission setting (p=0.35) which is in line with the reviewer’s comment. This is highlighted in the revised manuscript. 

Comment #18: 5. The epidemiological criteria for stratifying districts into low and high was not not indicated. 

Response: Thanks for the comment, this has been clarified in the revised manuscript. We used the ministry of health categorization of malaria transmission settings for identification of areas of high and low malaria transmission in the country. From the Ministry of health classification, we selected Tororo and Apac for high; then Mbarara and Kabale for low malaria transmission setting. 

Comment #19: 6. overall, the methods and the data generated, and the evidence reported do not adequately support the conclusion for such an important important intervention.

Response: Thanks for the comment, we followed the standard criteria by WHO/HAI for assessment of availability and price of medicines. In the survey areas (districts) all the drug outlets were considered for inclusion. Additionally, data was collected in settings of the country that represent areas with potentially low and high demand for subsidized ACTs. Addressing all the corrections as requested by the reviewers has further improved the manuscript.

---

## [Decision Letter · Decision Letter 1]

5 Feb 2024

PONE-D-23-37429R1Copayment mechanism in Uganda: Availability, market share and price of quality assured artemisinin-based combination therapies in private drug outletsPLOS ONE

Dear Dr. Ocan,

Thank you for submitting your manuscript to PLOS ONE. After careful consideration, we feel that it has merit but does not fully meet PLOS ONE’s publication criteria as it currently stands. Therefore, we invite you to submit a revised version of the manuscript that addresses the points raised during the review process.

We look forward to receiving your revised manuscript.

Kind regards,

Bosco Bekiita Agaba, PhD

Academic Editor

PLOS ONE

Additional Editor Comments (if provided):

The study aims to assess the availability, price, and market share of quality assured artemisinin-based combination therapies (QAACT) in the private drug outlets in low and high malaria transmission settings in Uganda” The authors conducted the study using a cross-sectional design covering and drawing their conclusions based on twenty-eight (28) drug outlets covered.

Overall, it is a useful study as it brings out data that could inform case management but also feeds into the overall ACTs resistance management plan.

Having read the concerns raised by the reviewers who are both subject matter experts at the WHO global malaria program and in my own view, there are some issues of methodological nature which should further improve the paper if addressed.

1. the ACTs procured under the subsidized/copayment mechanism are not distributed to the 4 districts covered by the survey alone but to the entire country of 140 + districts). In my opinion, it a big source of bias to base on four purposely/conveniently picked districts to report and make conclusion on the impact of a decade long program. This has also been raised by expert reviewer 2. It would generally be a good and informative study if reported findings from the 4 four the four study districts including a disclaimer that these findings do not reflect the effects of copayments in Uganda but in the study districts.

2. As raised earlier The supply chain system (first line buyers, retailers etc) through which these subsidized/copayment ACTs are distributed entirely relies on market forces, demand creation, etc. Because of these forces and dynamics, there are variations in the market share of these drugs between areas/regions within a country. Eg if you have more distributors moving to same direction you are likely to find more copaid drugs there compared to the rest. The best suitable method to answer the question would therefore entail sampling of outlets in a way that is representative and spread across wide coverage of the country. In their response, the authors mentioned that they samples from the various epidemiological endemicity but that does not address the risk of selection of bias due to an up-hazard distribution system that may cause over supply of co-paid drugs in some areas compared to the others. The up-hazard distribution and selective and non-probability convenient selection of districts makes generalization of this study difficult.

3. Reviewer 2 raises a very important aspect about the changing malaria epidemiology in Africa (Uganda inclusive)- malaria epidemiology is highly dynamic and transmission is highly heterogenous and this complicates epidemiological stratification into high, Moderate and low transmission merely based on parasite prevalence/slide positivity rates alone. Areas that were originally high are now low and vice versa and this phenomenon is happening in most countries in Africa. Accurate Epidemiological stratification is based on a composite factor that combines entomological inoculation rates (EIRs), parasite rates & positivity rates, etc. As raised by reviiwer 2 it nt clear how the decision stratify into high and low transmission was reached- raising questions on whether study areas such as Tororo and Apac are correctly epidemiologically placed. what parameters were based arrive at the transmission levels.

4. Overall, this is a very useful that is publishable- however the findings should be tailored to the 4 study districts. Like wise, conclusions can only be true for the study areas (would be better to retain the current title but add something like "..........; a case of four districts in Uganda"

Reviewers' comments:

Reviewer's Responses to Questions

**Comments to the Author**

1. If the authors have adequately addressed your comments raised in a previous round of review and you feel that this manuscript is now acceptable for publication, you may indicate that here to bypass the “Comments to the Author” section, enter your conflict of interest statement in the “Confidential to Editor” section, and submit your "Accept" recommendation.

Reviewer #1: (No Response)

Reviewer #2: (No Response)

2. Is the manuscript technically sound, and do the data support the conclusions?

Reviewer #1: Yes

Reviewer #2: Partly

3. Has the statistical analysis been performed appropriately and rigorously? 

Reviewer #1: Yes

Reviewer #2: Yes

4. Have the authors made all data underlying the findings in their manuscript fully available?

Reviewer #1: Yes

Reviewer #2: Yes

5. Is the manuscript presented in an intelligible fashion and written in standard English?

Reviewer #1: Yes

Reviewer #2: Yes

6. Review Comments to the Author

Reviewer #1: General:

Well written paper which gives a context for the improvement to access to quality assured ACTs across all sectors of the health system, particularly the private sector. However, a few areas (indicated below) would need minor adjustments to give the reader a true context of the situation and thus help in addressing the challenges this study has identified.

Introduction:

1. Though Co-payment mechanism is over 10 years in Uganda, the context has changed overtime. It would be helpful to give a summary of the Co-payment mechanism as at the time of the study, e.g. which antimalarial medicines in the national guidelines is a beneficiary of the scheme. A sentence in the introduction reads "The approved quality assured ACTs in Uganda include Artemether/lumefantrine, Artesunate/Amodiaquine, Dihydroartemisinin/piperaquine, and Artesunate/Pyronaridine", is not factual. Please clarify, is there dihydroartemisin/piperaquine and artesunate/pyronaridine with the "Green leaf" logo available. So though it is correct that all the medicines indicated in your statement are approved for use in Uganda, but not all of them benefit from co-payment scheme presently. Please clarify and modify the paper and conclusion accordingly.

Results:

1. Price of antimalarial agents in drug outlets in Uganda: - Some of the prices stated in the text does not match what is given in Table 2. Cross-check for consistency.

2. Price mark-ups on quality assured artemisinin-based combination therapies (QAACTs) in drug outlets in Uganda:-

What factors determine the final price? What is the buying price of the QAACT from first line buyers? Are the first line buyers maintaining the recommended selling prices to the pharmacies and drug shops?

Conclusion:

The conclusion that – “The results of our study indicate low prevalence of QAACTs in private drug outlets despite the implementation of private sector copayment mechanism in Uganda” is devoid of any contextual factors as to the finding. If these were collected as part of the study, it would enhance the strength and usefulness of the study results if they can be added to the paper.

Reviewer #2: This paper addresses a topical issue on access to ACT therapy in Uganda. I however have some comments that need to be addressed.

Major

1) In the background it is stated that that malaria accounts for 30-50% OPD attendances and 15-20 Hospital admissions. These data are almost 3 decades old. Is it really true that with recent interventions these statics have not changed?Secondly with IRS and LLINS implementation, is Apac and Tororo still ranked as high transmission areas. Please provide recent evidence to justify this stratification. This paper also misses out major papers on this subject in Uganda e.g Talisuna et al 2012, Gunther et al 2014

2. A second major limitation of this study is the sample size estimation of the number of the districts selected compared to the total number of districts in Uganda . As of December 2023, Uganda had close to 135 districts, so to conduct a study in 4 districts and then generalize the findings to 135 districts is Epidemiological injustice

3. The title of the study should be "Co-payment mechanism in Select District in Uganda: Availability, market share and price of quality assured artemisinin-based combination therapies in private drug outlets,

4. The Generalised conclusion both in the abstract and the main text needs to b revised what was studies here does surely not represent Uganda as close to 30 districts should have been sample

Minoe comments

Data us plural so the paper should be revised to reflect that " Data were analyzed using instead of data was analyzed using

7. PLOS authors have the option to publish the peer review history of their article (what does this mean?). If published, this will include your full peer review and any attached files.

Reviewer #1: **Yes: **Peter Olumese

Reviewer #2: **Yes: **Dr Ambrose Otau Talisuna, MBChB, Msc PhD

---

## [Author Response · Author response to Decision Letter 1]

4 Mar 2024

RESPONSE TO REVIEWER’S COMMENTS OF MANUSCRIPT PONE-D-23-37429R1

Title: Copayment mechanism in Uganda: Availability, market share and price of quality assured artemisinin-based combination therapies in private drug outlets 

Editor’s comment: The study aims to assess the availability, price, and market share of quality assured artemisinin-based combination therapies (QAACT) in the private drug outlets in low and high malaria transmission settings in Uganda” The authors conducted the study using a cross-sectional design covering and drawing their conclusions based on twenty-eight (28) drug outlets covered.

Overall, it is a useful study as it brings out data that could inform case management but also feeds into the overall ACTs resistance management plan.

Having read the concerns raised by the reviewers who are both subject matter experts at the WHO global malaria program and in my own view, there are some issues of methodological nature which should further improve the paper if addressed.

Response: Thanks for the feedback, we have addressed the additional comments from both the editor and the reviewer’s which has helped further improve the manuscript. 

Editor’s comment #1: the ACTs procured under the subsidized/copayment mechanism are not distributed to the 4 districts covered by the survey alone but to the entire country of 140 + districts). In my opinion, it a big source of bias to base on four purposely/conveniently picked districts to report and make conclusion on the impact of a decade long program. This has also been raised by expert reviewer 2. It would generally be a good and informative study if reported findings from the 4 four the four study districts including a disclaimer that these findings do not reflect the effects of copayments in Uganda but in the study districts.

Response: This has been noted, and its true the subsidized ACTs are distributed to the entire country. We have adjusted the write up in the manuscript to reflect this and included in the limitation and conclusion of the study. 

Editor’s comments #2: As raised earlier The supply chain system (first line buyers, retailers etc) through which these subsidized/copayment ACTs are distributed entirely relies on market forces, demand creation, etc. Because of these forces and dynamics, there are variations in the market share of these drugs between areas/regions within a country. Eg if you have more distributors moving to same direction you are likely to find more copaid drugs there compared to the rest. The best suitable method to answer the question would therefore entail sampling of outlets in a way that is representative and spread across wide coverage of the country. In their response, the authors mentioned that they samples from the various epidemiological endemicity but that does not address the risk of selection of bias due to an up-hazard distribution system that may cause over supply of co-paid drugs in some areas compared to the others. The up-hazard distribution and selective and non-probability convenient selection of districts makes generalization of this study difficult.

Response: Thanks for the comment, we have adjusted the title and conclusion to limit the findings to only the four (4) districts where data was collected. 

Editor’s comments #3: Reviewer 2 raises a very important aspect about the changing malaria epidemiology in Africa (Uganda inclusive)- malaria epidemiology is highly dynamic, and transmission is highly heterogenous, and this complicates epidemiological stratification into high, Moderate and low transmission merely based on parasite prevalence/slide positivity rates alone. Areas that were originally high are now low and vice versa and this phenomenon is happening in most countries in Africa. Accurate Epidemiological stratification is based on a composite factor that combines entomological inoculation rates (EIRs), parasite rates & positivity rates, etc. As raised by reviiwer 2 it nt clear how the decision stratify into high and low transmission was reached- raising questions on whether study areas such as Tororo and Apac are correctly epidemiologically placed. what parameters were based arrive at the transmission levels.

Response: We followed the Ministry of Health stratification of the malaria transmission intensity in the country. According to a publication by WHO in 2023 Uganda has the world’s highest malaria incidence rate of 478 cases per 1,000 population per year. It is also the leading cause of sickness and death in Uganda and is responsible for up to 40 percent of all outpatient visits, 25 percent of hospital admissions and 14 percent of all hospital deaths. The malaria death rate in Uganda is estimated to be between 70,000- 100,000 deaths per year (WHO malaria country report, 2023). Malaria transmission is stratified into four levels based on malaria cases, ‘high’ (> 499 cases per 1000 population/year), ‘moderate’ (200 to 499 cases per 1000 population per year), ‘low’ level 2 (50 to 199 cases per 1000 population per year), ‘very low’ level 1 (1-49 cases per 1000 population per year) (Zalwango et al., 2022).

The malaria test positivity rate (TPR) in Tororo district decreased from 60% (2012 to 2013) to 31% during final period of Actellic indoor residual spraying (IRS) and LLIN (June 2016 to December 2019) (Mpimbaza et al., 2020; Nankabirwa et al., 2022). Although there has been remarkable decrease in malaria prevalence in Tororo and Apac districts, these gains remain fragile due to inconsistency in implementation of the malaria vector control measures (Nankabirwa et al., 2022). Previous studies in Uganda (Esptein et al., 2023; Epstein et al., 2022) reported a resurgence in malaria burden to pre-IRS levels despite sustained IRS across sentinel surveillance sites. Therefore, despite implementation of various control interventions, malaria remains a public health concern in historically high burden districts in North-Eastern Uganda like Apac and Tororo districts where the current study was conducted. 

Uganda Bureau of Statistics (UBOS) and ICF. 2018. Uganda Demographic and Health Survey 2016. Kampala, Uganda and Rockville, Maryland, USA: UBOS and ICF.

Nankabirwa JI, Bousema T, Blanken SL, Rek J, Arinaitwe E, Greenhouse B, et al. (2022) Measures of malaria transmission, infection, and disease in an area bordering two districts with and without sustained indoor residual spraying of insecticide in Uganda. PLoS ONE 17(12): e0279464

Mpimbaza, A., Sserwanga, A., Rutazaana, D. et al. Changing malaria fever test positivity among paediatric admissions to Tororo district hospital, Uganda 2012–2019. Malar J 19, 416 (2020). 

Epstein A, Maiteki-Sebuguzi C, Namuganga JF, Nankabirwa JI, Gonahasa S, Opigo J, Staedke SG, Rutazaana D, Arinaitwe E, Kamya MR, Bhatt S, Rodríguez-Barraquer I, Greenhouse B, Donnelly MJ, Dorsey G. Resurgence of malaria in Uganda despite sustained indoor residual spraying and repeated long lasting insecticidal net distributions. PLOS Glob Public Health. 2022 Sep 7;2(9):e0000676. 

Epstein A, Namuganga JF, Nabende I, et al. Mapping malaria incidence using routine health facility surveillance data in Uganda. BMJ Global Health 2023;8:e011137

Editor’s comments #4: Overall, this is a very useful that is publishable- however the findings should be tailored to the 4 study districts. Like wise, conclusions can only be true for the study areas (would be better to retain the current title but add something like "..........; a case of four districts in Uganda"

Response: Thanks for the comment, the title has been revised to reflect the fact that the study was conducted in four (4) districts. 

Reviewer’s comment #1: General:

Well written paper which gives a context for the improvement to access to quality assured ACTs across all sectors of the health system, particularly the private sector. However, a few areas (indicated below) would need minor adjustments to give the reader a true context of the situation and thus help in addressing the challenges this study has identified.

Response: Thanks for the comment and have addressed this in the revised manuscript. 

Reviewer’s comment #2. Though Co-payment mechanism is over 10 years in Uganda, the context has changed overtime. It would be helpful to give a summary of the Co-payment mechanism as at the time of the study, e.g. which antimalarial medicines in the national guidelines is a beneficiary of the scheme. A sentence in the introduction reads "The approved quality assured ACTs in Uganda include Artemether/lumefantrine, Artesunate/Amodiaquine, Dihydroartemisinin/piperaquine, and Artesunate/Pyronaridine", is not factual. Please clarify, is there dihydroartemisin/piperaquine and artesunate/pyronaridine with the "Green leaf" logo available. So though it is correct that all the medicines indicated in your statement are approved for use in Uganda, but not all of them benefit from co-payment scheme presently. Please clarify and modify the paper and conclusion accordingly.

 Response: As indicated in the manuscript, the approved quality assured ACTs are as indicated in the figure. 

https://onehealthtrust.org/publications/archive/affordable-medicines-facility-malaria-ugandas-perspective/

Reviewer’s comment #3. Price of antimalarial agents in drug outlets in Uganda: - Some of the prices stated in the text does not match what is given in Table 2. Cross-check for consistency.

Response: Thanks for the observation, this has been rectified in Table 2 of the revised manuscript. 

Reviewer’s comment #4. Price mark-ups on quality assured artemisinin-based combination therapies (QAACTs) in drug outlets in Uganda:-

What factors determine the final price? What is the buying price of the QAACT from first line buyers? Are the first line buyers maintaining the recommended selling prices to the pharmacies and drug shops?

Response: We did not assess the buying price of first line buyers. However, in a separate qualitative study (Manuscript under review) we explored and reported on the determinants of final QAACT price in the private drug outlets in selected districts in Uganda. In this study we focused on access to QAACT by the malaria patients in private drug outlets. We also report on concordance between dispensing price of QAACT in private drug outlets and the recommended retail prices as per the Ministry of Health. 

Reviewer’s comment #5: The conclusion that – “The results of our study indicate low prevalence of QAACTs in private drug outlets despite the implementation of private sector copayment mechanism in Uganda” is devoid of any contextual factors as to the finding. If these were collected as part of the study, it would enhance the strength and usefulness of the study results if they can be added to the paper.

Response: We collected contextual factors and performed multivariable logistic regression and the results are reported in the manuscript. 

Reviewer’s comment #6: This paper addresses a topical issue on access to ACT therapy in Uganda. I however have some comments that need to be addressed.

Response: We are grateful for the comments, addressing these comments have improved our manuscript. 

Reviewer’s comment #7: In the background it is stated that that malaria accounts for 30-50% OPD attendances and 15-20 Hospital admissions. These data are almost 3 decades old. Is it really true that with recent interventions these statics have not changed?Secondly with IRS and LLINS implementation, is Apac and Tororo still ranked as high transmission areas. Please provide recent evidence to justify this stratification. This paper also misses out major papers on this subject in Uganda e.g Talisuna et al 2012, Gunther et al 2014

Response: Malaria contributing to 30-50% of OPD attendances and 15-20 Hospital admissions is within the range of the current observed malaria burden in the country. According to a publication by WHO in 2023 Uganda has the world’s highest malaria incidence rate of 478 cases per 1,000 population per year. It is also the leading cause of sickness and death in Uganda and is responsible for up to 40 percent of all outpatient visits, 25 percent of hospital admissions and 14 percent of all hospital deaths (WHO, malaria report, 2023; MoH, 2024). 

The malaria death rate in Uganda is estimated to be between 70,000 and 100,000 deaths per year (WHO malaria country report, 2023). Malaria transmission which is stratified into four levels based on malaria cases, ‘high’ (> 499 cases per 1000 population/year), ‘moderate’ (200 to 499 cases per 1000 population per year), ‘low’ level 2 (50 to 199 cases per 1000 population per year), ‘very low’ level 1 (1-49 cases per 1000 population per year) (Zalwango et al., 2022). 

The malaria test positivity rate (TPR) in Tororo district decreased from 60% (2012 to 2013) to 31% during final period of Actellic indoor residual spraying (IRS) and LLIN (June 2016 to December 2019) (Mpimbaza et al., 2020; Nankabirwa et al., 2022). Although there has been remarkable decrease in malaria prevalence in Tororo and Apac districts, these gains remain fragile due to inconsistency in implementation of the malaria vector control measures (Nankabirwa et al., 2022). Previous studies in Uganda (Esptein et al., 2023; Epstein et al., 2022; Talisuna et al., 2012) reported a resurgence in malaria burden to pre-IRS levels despite sustained IRS across sentinel surveillance sites. Therefore, despite implementation of various control interventions, malaria remains a public health concern in historically high burden districts in North-Eastern Uganda like Apac and Tororo districts where the current study was conducted. Based on the current evidence on malaria burden in the study districts, we have revised the categorization of Apac and Tororo as having moderate-to-high malaria transmission (Talisuna et al., 2012; Esptein et al., 2022). 

Talisuna A, Adibaku S, Dorsey G, Kamya MR, Rosenthal PJ. Malaria in Uganda: challenges to control on the long road to elimination. II. The path forward. Acta Trop. 2012 Mar;121(3):196-201.

Zalwango, M.G., Zalwango, J.F., Kadobera, D. et al. Evaluation of malaria outbreak detection methods, Uganda, 2022. Malar J 23, 18 (2024).

Nankabirwa JI, Bousema T, Blanken SL, Rek J, Arinaitwe E, Greenhouse B, et al. (2022) Measures of malaria transmission, infection, and disease in an area bordering two districts with and without sustained indoor residual spraying of insecticide in Uganda. PLoS ONE 17(12): e0279464

Mpimbaza, A., Sserwanga, A., Rutazaana, D. et al. Changing malaria fever test positivity among paediatric admissions to Tororo district hospital, Uganda 2012–2019. Malar J 19, 416 (2020). 

Epstein A, Maiteki-Sebuguzi C, Namuganga JF, Nankabirwa JI, Gonahasa S, Opigo J, Staedke SG, Rutazaana D, Arinaitwe E, Kamya MR, Bhatt S, Rodríguez-Barraquer I, Greenhouse B, Donnelly MJ, Dorsey G. Resurgence of malaria in Uganda despite sustained indoor residual spraying and repeated long lasting insecticidal net distributions. PLOS Glob Public Health. 2022 Sep 7;2(9):e0000676. 

Uganda Bureau of Statistics (UBOS) and ICF. 2018. Uganda Demographic and Health Survey 2016. Kampala, Uganda and Rockville, Maryland, USA: UBOS and ICF.

Reviewer’s comment #8: A second major limitation of this study is the sample size estimation of the number of the districts selected compared to the total number of districts in Uganda. As of December 2023, Uganda had close to 135 districts, so to conduct a study in 4 districts and then generalize the findings to 135 districts is Epidemiological injustice

Response: Thanks for the comment, the title of the study, conclusion and overall write-up has been adjusted to reflect the fact that data was collected from only four districts in the country. However, the four districts cover diverse malaria transmission settings in the country. 

Reviewer’s comment #9. The title of the study should be "Co-payment mechanism in Select District in Uganda: Availability, market share and price of quality assured artemisinin-based combination therapies in private drug outlets,

Response: Thanks, this has been incorporated in the revised manuscript. 

Reviewer’s comment #10. The Generalised conclusion both in the abstract and the main text needs to b revised what was studies here does surely not represent Uganda as close to 30 districts should have been sample

Response: Thanks for the comment, we have restricted the conclusion in the revised manuscript to the four (4) districts from which the data was collected. 

Reviewer’s comment #11: Data us plural so the paper should be revised to reflect that " Data were analyzed using instead of data was analyzed using. 

Response: Thanks, this has been corrected in the revised manuscript.

---

## [Editor Report · Decision Letter 2]

12 Mar 2024

Copayment mechanism in selected districts of Uganda: Availability, market share and price of quality assured artemisinin-based combination therapies in private drug outlets.

PONE-D-23-37429R2

Dear Dr. Ocan Moses

We’re pleased to inform you that your manuscript has been judged scientifically suitable for publication and will be formally accepted for publication once it meets all outstanding technical requirements.

Kind regards,

Bosco Bekiita Agaba, PhD

Academic Editor

PLOS ONE

Additional Editor Comments (optional):

All comments have been addressed and the paper read well.
---

## [Editor Report · Acceptance letter]

18 Mar 2024

PONE-D-23-37429R2 

PLOS ONE

Dear Dr. Ocan, 

I'm pleased to inform you that your manuscript has been deemed suitable for publication in PLOS ONE. Congratulations! Your manuscript is now being handed over to our production team.

Kind regards, 

on behalf of

Dr. Bosco Bekiita Agaba 

Academic Editor

PLOS ONE